# Targeting ALK in Cancer: Therapeutic Potential of Proapoptotic Peptides

**DOI:** 10.3390/cancers11030275

**Published:** 2019-02-26

**Authors:** Arthur Aubry, Stéphane Galiacy, Michèle Allouche

**Affiliations:** 1Lunenfeld Tanenbaum Research Institute, Mount Sinai Hospital, Toronto, ON M5G 1X5, Canada; arthur-aubry@hotmail.fr; 2Department of Laboratory Medicine and Pathobiology, University of Toronto, Toronto, ON M5S 1A1, Canada; 3INSERM, UDEAR, UMR1056, F-31300 Toulouse, France; stephane.galiacy@inserm.fr; 4University of Toulouse, F-31300 Toulouse, France; 5Department of Ophthalmology, Toulouse University Hospital, F-31300 Toulouse, France

**Keywords:** anaplastic lymphoma kinase, ALK, tyrosine kinase, dependence receptor, proapoptotic peptides, tyrosine kinase inhibitor, anaplastic large cell lymphoma, non-small-cell lung cancer, neuroblastoma, targeted therapy

## Abstract

ALK is a receptor tyrosine kinase, associated with many tumor types as diverse as anaplastic large cell lymphomas, inflammatory myofibroblastic tumors, breast and renal cell carcinomas, non-small cell lung cancer, neuroblastomas, and more. This makes ALK an attractive target for cancer therapy. Since ALK–driven tumors are dependent for their proliferation on the constitutively activated ALK kinase, a number of tyrosine kinase inhibitors have been developed to block tumor growth. While some inhibitors are under investigation in clinical trials, others are now approved for treatment, notably in ALK-positive lung cancer. Their efficacy is remarkable, however limited in time, as the tumors escape and become resistant to the treatment through different mechanisms. Hence, there is a pressing need to target ALK-dependent tumors by other therapeutic strategies, and possibly use them in combination with kinase inhibitors. In this review we will focus on the therapeutic potential of proapoptotic ALK-derived peptides based on the dependence receptor properties of ALK. We will also try to make a non-exhaustive list of several alternative treatments targeting ALK-dependent and independent signaling pathways.

## 1. Introduction

Anaplastic Lymphoma Kinase (ALK) was first discovered in 1994 as part of the nucleophosmin (NPM)-ALK fusion resulting from the highly recurrent (2;5)(p23;q35) translocation in anaplastic large cell lymphomas (ALCL) [1,2]. After that, this NPM-ALK fusion and other ALK rearrangements, including gene translocations or inversions with different fusion partners, were identified, both in hematopoietic and non-hematopoietic solid cancers. For instance, ALK translocations or inversions are frequent in inflammatory myofibroblastic tumors (IMT) (50–60%), and 4–8% of non-small cell lung cancer (NSCLC) carry an echinoderm microtubule-associated protein-like 4 (EML4)-ALK fusion due to an (2)(p21;p23) inversion. ALK translocations are also present in diffuse large B cell lymphomas, breast, and renal cell carcinomas at low frequency [3,4,5]. The fusion proteins arising from these cytogenetic rearrangements associate the N-terminal part of the partner, usually containing a dimerization domain, and the entire intracellular portion of ALK, which includes its tyrosine kinase domain. Once expressed, the fusion proteins dimerize, causing aberrant constitutive autophosphorylation and activation of the ALK kinase, which promotes uncontrolled cell proliferation and survival [6].

The full length ALK receptor cDNA was isolated in 1997, and codes for a 1620 amino acid transmembrane receptor tyrosine kinase of the insulin receptor superfamily. ALK is almost exclusively expressed in the central and peripheral nervous system during development. In the adult, a low expression of *ALK* mRNA has been reported in a few restricted zones of the brain, in the small intestine (likely within the intestinal peripheral nervous system ganglia), and in testis [2,7,8]. Expression of the full length ALK receptor was also observed in neuroblastoma, a tumor of embryonic origin affecting the peripheral nervous system. The ALK kinase in some neuroblastomas is activated as a result of gain-of-function mutations, *ALK* gene amplification, or copy number increase [9,10,11,12,13]. Other solid tumors expressing the full length ALK receptor include rhabdomyosarcomas, glioblastomas, melanomas, and retinoblastomas [14]. However, an oncogenic role of ALK in these tumors has not been clearly demonstrated yet. 

The physiological role of ALK has been only partially elucidated in model animals. In *Drosophila melanogaster*, DAlk binds a ligand named jelly belly (jeb). Lack of either *DAlk* or *jeb* leads to an abnormal development of the visceral mesoderm [15,16,17,18]. DAlk is also expressed in the fly nervous system [19]. In the fly developing visual system, the DAlk/jeb couple plays a central role in neuronal circuit assembly [20]. Moreover, in *Caenorhabditis elegans* an *ALK* homologue expressed in the nervous system was proposed to regulate synapse differentiation at neuromuscular junctions [21]. However, in vertebrates, ALK has remained an orphan receptor for many years. Early reports proposed pleiotrophin (PTN) and midkine, two related heparin-binding factors, as ligands for ALK [6]. However, direct binding to ALK was not reproduced by other groups [22,23], and further evidence suggested it could be mediated by heparin [24]. In addition, PTN and midkine are genuine ligands for the transmembrane receptor tyrosine phosphatase β/ζ (RPTPβ/ζ), which they inactivate. In glioblastoma cells, Perez-Pinera et al. showed RPTPβ/ζ was active in the absence of ligand and could dephosphorylate ALK, suggesting it could play a regulatory role in ALK signaling [25].

During evolution, *ALK* was likely duplicated, as it has a strong homology with *LTK*, a gene also expressed in vertebrates but not in *C. elegans* or *D. melanogaster*. ALK and LTK share similarities in the kinase domain, as well as a glycine-rich region in their extracellular domains [7,8]. Zhang et al. recently screened a large collection of extracellular proteins individually expressed and secreted by numerous human tissues for their ability to activate LTK. They identified two novel human secreted proteins as ligands for LTK, the family with sequence similarity (FAM) 150A and FAM150B [26]. FAM150B (also called Augmentor-α) could act as a universal, dual-specific ligand for both ALK and LTK, whereas FAM150A (or Augmentor-β) might have a higher affinity for LTK [27]. Palmer’s group showed that both FAM150A and FAM150B were not only able to bind and activate human wild type ALK, but also constitutively active ALK mutants from neuroblastoma, resulting in a further enhanced activity or ‘superactivation’ [28].

These secreted molecules have been named ALK and LTK ligands (ALKAL), as approved by HUGO Gene Nomenclature Committee. Thus, FAM150A is the same as ALKAL1 and Augmentor-β, while FAM150B is the same as ALKAL2 and Augmentor-α. Biochemical characterization of purified ALKAL2 demonstrated the presence of two conserved intramolecular disulfide bridges important for activation of the receptors, whereas an intermolecular disulfide bridge (mediating ligand dimerization) appeared dispensable for this function [29]. 

Recently, two groups demonstrated that zebrafish Ltk (DrLtk), which is more similar to human ALK than to LTK in sequence and domain structure, controls the development of iridophores (zebrafish-specific pigment cells) [30,31]. It is worthy to note that both zebrafish iridophores and human neuroblastoma, an embryonic tumor, derive from the neural crest, showing the importance of ALK and DrLtk in development. However, in humans, the physiological role of ALK remains elusive. De Pontual et al. reported two cases of germline gain-of-function mutations of *ALK*, associating severe defects of the central nervous system together with congenital neuroblastoma. Their observations illustrate the role of activated ALK kinase in both tumor predisposition and normal development of the nervous system, and shed light on the pleiotropic role of ALK in humans [32].

## 2. ALK Signaling

### 2.1. ALK Proliferative and Pro-Survival Cell Signaling 

Similar to other tyrosine kinase receptors localized at the cell surface, activation of full length ALK requires ligand binding to induce ALK dimerization and autophosphorylation of the kinase, which in turn triggers signaling pathways leading to proliferation, differentiation, or migration [33]. For example, neuroblastoma cells treated with ALK-activating monoclonal antibodies or with FAM150 ligands promote cell proliferation or differentiation while activating the ERK1/2 and ERK5 pathways [22,28,34,35]. The proliferative function of ALK is also achieved by constitutive activation, due to gene amplification or gain-of-function mutations, which are prevalent in approximately 10% neuroblastomas [12,36] (Figure 1A). 

In contrast, ALK fusion proteins expressed in tumors, such as NPM-ALK in ALCL or EML4-ALK in NSCLC, are not expressed at the cell surface and remain intracellular. The mechanism of activation of ALK fusion proteins requires, in most cases, a dimerization or multimerization domain provided by the fusion partner [37,38] (Figure 1A). Dimerized, kinase-active EML4-ALK and NPM-ALK are located in the cytoplasm, whereas NPM-ALK also localizes to the nucleus and nucleolus, where it forms heterodimers with NPM1, thus, it is not activated in these compartments [37,39]. 

Recently, another set of three constitutively activated ALK truncated isoforms of 58–61 kDa was discovered in a subset of human melanomas (11% of cases). These novel proteins, which are essentially composed of the intracellular kinase domain of ALK, arise from an alternative transcription initiation (*ALK*^ATI^), and localize both in the cytoplasm and cell nucleus. Expression of *ALK*^ATI^ allows growth factor-independent proliferation of murine BaF3 cells and promotes tumor formation from transfected NIH-3T3 cells in a mouse model. Similar to other ALK isoforms, ALK^ATI^ proteins are activated by dimerization and autophosphorylation [40,41] (Figure 1A). 

Signaling pathways triggered by active ALK have been extensively studied in the context of NPM-ALK expressing ALCL. They trigger cell proliferation and prevent apoptosis [39]. These pathways are also functioning in other ALK fusion tumors [38]. The four main signaling pathways downstream of kinase-activated ALK are depicted in Figure 1A. They include phosphoinositol-3 kinase (PI3K)/Akt, signal transducer and activator of transcription 3 (STAT3), phospholipase Cγ (PLCγ), and extracellular signal-regulated kinases (ERK) 1/2 and 5. The preferentially used pathways vary depending on the ALK isoform (full length or truncated, wild type or mutated, fused with a protein partner, etc.), and subcellular localization [35,40,42,43]. The nuclear ALK^ATI^ isoforms are also phosphorylated, and have been shown to induce chromatin structural modifications, however, their substrates and function in the nucleus need to be further clarified [41]. The type of ALK-expressing tumor (ALCL, IMT, NSCLC, neuroblastoma, melanoma, etc.) and possible concomitant activation of oncogenes, such as MYCN or K-RAS in neuroblastomas, for example, are also determinant for the choice of one particular signaling pathway by tumor cells [42,44]. 

### 2.2. ALK Proapoptotic Signaling

Our group previously demonstrated ALK is a dependence receptor [23,45]. The dependence receptor family includes RTKs like MET, RET, TRKA, TRKC, insulin receptor, IGF1R, and ALK, as well as transmembrane receptors with different structures and signal transduction such as, for instance, DCC, UNC5H, neogenin, or PTCH1 [46]. Dependence receptors are characterized by the activation of radically opposite intracellular signaling pathways depending on the presence or absence of their respective ligands, or any molecule mimicking them (such as activating antibodies). In the presence of their ligand (or equivalent), dependence receptors induce cell survival and proliferation (or in some cases differentiation or migration), whereas in the absence of ligand, and if the cell is confronted with environmental or genotoxic stress, these receptors actively trigger apoptosis, as opposed to conventional receptors that would not. The proapoptotic effect depends on caspase-mediated cleavage of the receptor, which exposes an addiction/dependence domain (ADD) that is pivotal for enhancing the apoptotic cascade [47,48].

Note that many ligands of dependence receptors are neurotrophins playing a crucial role during development of the nervous system [49]. A defective expression of these receptors or their ligands can lead to severe developmental defects [50]. Alternatively, overexpressed or gain-of-function mutated dependence receptors can play a role in cancer as oncogenes, as shown for ALK [45]. ALK dependence receptor proapoptotic signaling is illustrated in Figure 1B.

In summary, ALK is abundantly expressed and oncogenic in multiple cancer types affecting children (ALCL and neuroblastomas) and adults (NSCLC and other solid tumors), while it is absent from the majority of normal adult tissues. These characteristics make ALK an attractive candidate for cancer therapy. ALK has pro- and recently identified anti-proliferative functions, depending on the context. This duality unravels multiple possible strategies to block ALK-dependent cancers: (1) inhibiting the ALK proliferative signaling with direct targeting of ALK or indirect targeting of downstream effectors; (2) activating the ALK proapoptotic signaling.

## 3. ALK Tyrosine Kinase Inhibitors

### 3.1. First Generation ALK Tyrosine Kinase Inhibitor: Crizotinib

Different approaches used to decrease or inactivate ALK, such as shRNAs, blocking antibodies, or loss-of-function mutants have shown that the growth of ALK-driven tumors is dependent on ALK expression level and kinase phosphorylation [22,36,37,51]. Because RNA knockdown techniques were not yet ready for clinical applications, an attractive strategy to kill oncogene-addicted cancer cells was to target the kinase with inhibitors. Small molecule tyrosine kinase inhibitors (TKIs) specifically targeting ALK have been developed by both academia and pharmaceutical companies [52,53,54,55,56]. Some of these molecules inhibit more than one tyrosine kinase. For example, crizotinib, the first ALK TKI approved by the FDA for treating patients, was initially described as a dual MET/ALK kinase inhibitor, and was later found to be active on ROS1-driven tumors [52,57]. In early clinical trials, crizotinib was shown to improve overall survival of patients with advanced ALK-positive NSCLC or ALCL [58,59]. Moreover, results from the PROFILE 1014 study revealed crizotinib was superior to standard first-line pemetrexed-plus-platinum chemotherapy in patients with previously untreated advanced ALK-positive NSCLC [60].

### 3.2. Next-Generation ALK Tyrosine Kinase Inhibitors 

Unfortunately, after an initial response to treatment with TKIs, tumor cell resistance invariably develops, leading to relapse. It is important to analyze the mechanisms of resistance in order to fight them. A frequent event in ALK-driven resistance is the appearance of secondary mutations in the kinase domain, whereas ALK gene amplification can also occur [61,62]. Interestingly, some sites of secondary mutations observed in relapses of crizotinib-treated tumors are often the same as those described in primary neuroblastomas [63]. To circumvent these mutations, second generation ALK inhibitors have been developed. Notably, ceritinib and alectinib have demonstrated efficacy in crizotinib-resistant patients, as well as first-line therapy in NSCLC [64]. Brigatinib is efficient in vitro on a number of ALK secondary mutations, and has achieved durable responses in ALK-rearranged NSCLC patients previously treated by crizotinib [65]. These new TKIs are particularly useful to treat brain metastases, as they can cross the blood–brain barrier [66,67,68]. Interestingly, after alternate treatments, a tumor may become sensitive again to crizotinib. Kashima et al. reported a patient with multiple brain metastases who was heavily treated with cytotoxic chemotherapy, and previously treated with crizotinib and alectinib, whose brain metastases regressed following rechallenge with crizotinib [69]. However, in some patients with ALK-positive NSCLC treated sequentially with several ALK TKI, the tumor or its metastases become resistant to second generation ALK inhibitors, due to the occurrence of compound mutations in the kinase domain of ALK [62], or to the activation of bypass signaling pathways (see below). Therefore, third generation ALK TKI were designed, with lorlatinib being a leader molecule. This TKI targets both ALK and ROS1, is more efficient on mutants than wild type ALK, and can cross the blood–brain barrier. Both preclinical assays and clinical trials are very promising [62,70,71]. Note that each TKI has a unique profile of activity against a distinct set of mutations. A recent review by Sharma et al. listed the different ALK TKI, their specificity for other kinases, the (completed or ongoing) clinical trials currently testing these inhibitors and their observed adverse effects [72]. Another study focused more specifically on clinical trials of ALK inhibitors in neuroblastoma [73].

Thus, in order to overcome resistance to treatment by TKIs, it appears necessary to design other tumor-targeting strategies, and possibly associate these different treatments with TKIs. Beside a higher and prolonged efficiency of the treatment, drug combination may allow use of lower doses of each component and potentially minimize drug-related adverse events. 

## 4. Overcoming Resistance to ALK TKI

### 4.1. Targeting ALK-Dependent Signaling Pathways

One way to counteract ALK-addicted tumor proliferation is to target ALK downstream signaling (Figure 1A). The importance of STAT3 signaling pathway has been widely demonstrated in NPM-ALK-dependent ALCL [74]. Moreover, STAT3 signaling in tumor cells can increase following treatment with TKIs in a feedback mechanism [75]. Thus, inhibiting STAT3 could represent a strategy to hamper both ALK-dependent proliferation and TKI resistance. Unfortunately, the currently available STAT3 inhibitors demonstrate a low specificity or high toxicity, and cannot be used in the clinic [76].

The PI3K and ERK/MEK pathways are also frequently stimulated in tumors bearing an activated ALK isoform. Inhibitors of these pathways have been tested in different experimental conditions. Separately, they have no effect, but an association of a PI3K and a MEK inhibitor can decrease proliferation and induce apoptosis of EML4-ALK bearing H3122 human lung cancer cells. However, this treatment did not achieve tumor regression of H3122 xenografts, nor of *EML4-ALK*-driven lung cancer in mice [77,78]. 

Another ALK targeting approach aims at stimulating its degradation by the ubiquitin–proteasome complex. Indeed, ALK fusion molecules such as NPM-ALK in ALCL or EML4-ALK in NSCLC are client proteins of Hsp90, which protects them from degradation. Several groups have shown Hsp90 inhibitors are active in vitro and in vivo to kill ALK-positive tumor cells. However, the antitumor effect in these models was less durable than with ALK inhibitors [77,79,80].

### 4.2. Targeting ALK-Independent Signaling Pathways

ALK-positive NSCLC treated with crizotinib become resistant with a median occurrence of approximately 10 months after treatment initiation. Even with the second/third generation ALK inhibitors, tumors become resistant [72]. One of the mechanisms allowing tumor cells to overcome ALK oncogene addiction involves the activation of bypass signaling pathways. Examples of such pathways include autocrine or paracrine activation of the EGFR, c-MET, c-KIT, PDGFR, and IGFR signaling [81,82,83]. De novo mutations of other oncogenes, such as EGFR or KRAS, can also occur in NSCLC. Concurrent ALK/KRAS co-alterations are associated with resistance to ALK TKI treatment [84]. Following relapse after an ALK inhibitor therapy, treatments targeted at these alternate pathways have been attempted with variable success (reviewed by Sharma et al. [72]). Preclinical studies would advise inhibition of at least one other pathway in addition to ALK, or to treat with dual kinase inhibitors [81,85].

### 4.3. Immunotherapy

The recent discovery that immune checkpoint inhibitors can be efficient in the clinic to fight advanced and metastasized cancers has brought a therapeutic revolution in the treatment of lung cancer. Indeed, a number of NSCLC express the PD-L1 antigen, and can be treated with anti-PD1 antibodies. Targeting the PD1/PD-L1 axis is particularly interesting both in NSCLC and ALCL, because ALK fusions upregulate PD-L1 expression via STAT3 as transcription factor [86,87,88].

### 4.4. Triggering ALK Proapoptotic Signaling with ALK-Derived Peptides

Our group demonstrated ALK is a dependence receptor, and, as such, has a proapoptotic function [23,45]. Using structure/function assay with various truncated forms of ALK, we mapped the ADD domain of ALK within the juxtamembrane intracytoplasmic region of the receptor (Figure 2). The ADD domain of ALK is 36 amino acids in length and has no homology with other known ADD or motifs linked to the apoptotic pathway, yet it is necessary for the proapoptotic function of ALK [47]. In a therapeutic perspective, we then tested the idea that this peptide sequence could be used as novel ALK-targeting agent, alone or in combination with ALK-directed TKI (Figure 2). Indeed, we found the synthetic 36 aa ALK-derived peptide (P36) that mimics the ADD domain of ALK was specifically inducing apoptosis in a caspase-dependent fashion in ALK-positive ALCL and ALK-expressing neuroblastoma cells, but not ALK-negative cancer cell lines and normal peripheral blood mononuclear cells [89]. 

An important feature of the synthetic peptide was N-myristoylation. This modification was necessary for cell penetration and biological activity of the peptide. Two P36-derived shorter peptides, as well as a cyclic peptide, also induced apoptosis. In order to identify protein partners from two responsive cell lines, surface plasmon resonance followed by mass spectrometry analysis of P36-interacting proteins was performed. Cross-analysis of these two “interactomes” yielded 16 identical or homologous proteins from Cost ALCL and SH-SY5Y neuroblastoma. Remarkably, these proteins were divided into two major biological functions. The first function involved direct or indirect interactions with the p53 gene/protein and/or p53-dependent signaling pathways. This seemed relevant, as we showed siRNA-mediated knockdown of p53 prevented cell death in ALK-positive tumor cells, indicating P36-induced apoptosis was a p53-dependent phenomenon [89].

The second top biological function of the P36 interactome was alternative splicing, as many hits were pre-mRNA splicing factors. Consistently, these factors are known as important regulators of apoptosis, controlling expression of many genes including caspases, BCL-2, and p53 family members, further arguing that P36 activated a proapoptotic network in which p53 signaling was central [90]. Changes in alternative splicing are common in cancer, and there are emerging evidences that targeting this regulatory mechanism has strong implications in tumorigenicity and therapy [91,92]. One hypothesis to explain P36-induced apoptosis would be that sequestration and/or depletion of splicing factors through interaction with P36 could impair the balance of antiapoptotic and proapoptotic signals, and thus decrease tumor cell survival due to increased apoptosis. 

## 5. Combined Therapies: The Future

As seen previously, acquired resistance to TKIs is a recurrent issue in cancer. Thus, attacking the tumor on two fronts at a time could be a strategy to overcome the resistance. 

### 5.1. Association of an ALK TKI and a Proapoptotic Peptide

#### 5.1.1. Peptides as Therapeutics

The interest in therapeutic peptides has increased in the past years, due to their relatively easy synthesis and low toxicity. Peptides have been used as therapy in medicine since the 1920s, when insulin isolated from animal sources was administered to diabetic patients. In the 1950s, the elucidation of DNA and protein sequence allowed the chemical synthesis of native hormone peptides. However, peptides are easily degraded by proteases in vivo. To optimize their therapeutic effect, it has been necessary to synthesize peptide analogs, or, more recently, small molecule drugs acting as peptidomimetics. In cancer, the knowledge acquired in the mechanisms of tumor cell death induced by chemotherapy or radiotherapy has led to the development of peptide drugs able to kill tumor cells by apoptosis (reviewed by Lau and Dunn [93] and Marqus et al. [94]). One significant approach aimed at targeting molecular actors of the mitochondrial pathway of apoptosis [95]. Mai et al. discovered that the antimicrobial peptide (KLAKLAK)2 coupled to a cell penetrating sequence was proapoptotic in murine and human tumor cell lines [96]. However, it lacked specificity, and needed to be engineered with tumor-targeting molecules [97]. Walensky et al. designed peptidomimetics of the BH3 domain from proaapoptotic members of the BCL-2 family to induce apoptosis of cancer cells [98]. For example, the BCL-2 inhibitor peptidomimetics ABT-199 (Venetoclax) is presently used in clinical trials in certain lymphomas, leukemia, or multiple myeloma (reviewed by Ashkenazi et al. [99]). 

#### 5.1.2. ALK-Derived Peptides Enhance TKI-Induced Apoptosis

We investigated whether the P36 peptide could cooperate with a TKI to kill ALK-addicted tumor cells (Figure 2). Indeed, we observed that a treatment combining P36 with the ALK-specific inhibitor crizotinib resulted in additive cytotoxicity in ALK-bearing tumor cells in vitro in two cell models, ALCL, and neuroblastoma [89]. Therefore, the proapoptotic activity of ALK-derived peptides, alone or in association with TKI, towards ALK-dependent tumor cells demonstrates their therapeutic potential. Eventually, in order to trigger tumor cell killing in vivo, the development of peptidomimetics and/or the vectorization of the peptides, for example using nanoparticles [100], would be helpful. 

#### 5.1.3. The Proapoptotic Activity of ALK-Derived Peptides Depends on p53

We have shown that a functional p53 was necessary for the proapoptotic action of P36. The p53 protein is a transcriptional activator, playing an important role in cell cycle control, DNA damage response, and apoptosis. Defective p53 could allow abnormal cells to proliferate, favoring the development of cancer. Murine double minute 2 (MDM2) is the main regulator of p53, and triggers its degradation via the ubiquitin–proteasome system. As many as 50% of all human tumors contain p53 mutants. However, the majority of ALCL and neuroblastomas harbor a wild type p53 [101,102]. In a series of ALCL tumors from patients, Rassidakis et al. noted an overexpression of both p53 and MDM2, suggesting p53 could be inactive [102]. Therefore, it could be interesting to reactivate p53 in ALK-positive tumors. 

### 5.2. Reactivation of p53 with an MDM2 Inhibitor

MDM2 is an E3 ubiquitin ligase that directly interacts and inhibits p53 in a polyubiquitination-dependent and -independent manner. The first mechanism involves downregulation of p53 via acting on p53 stability through polyubiquitination and proteasome degradation, whereas the second repressive mechanism is associated with inhibition of p53 transcriptional activation at the promoter of p53 target genes [103]. Therefore, reactivation of p53 function using an MDM2 inhibitor appears as an interesting therapy that could be associated with ALK-targeted TKI and ALK-derived proapoptotic peptides. In preclinical models of ALK-positive ALCL, Drakos et al. have shown that p53 reactivation by the MDM2 inhibitor nutlin-3a increased p53 concentration, enhanced apoptosis, and decreased tumorigenicity [104]. Another study investigated the ALK TKI ceritinib in combination with the MDM2 inhibitor CGM097 in ALK-mutated and p53 wild type neuroblastoma. These authors reported that both drugs act synergistically to inhibit proliferation of neuroblastoma in vitro and in xenograft models. This treatment arrested cell cycle and tumor cell proliferation [105]. These results suggest that associating an MDM2 inhibitor with P36 peptide might be a strategy to enhance tumor cell apoptosis, a hypothesis not yet tested, with a caveat that wild type p53 reactivation may sometimes lead to secondary oncogenic mutations [106]. 

### 5.3. Association of an ALK TKI and an Hsp90 Inhibitor

Hsp90 is a molecular chaperone, acting as a protection for its client proteins from proteasomal degradation. Many Hsp90 “clients” are signaling kinases critical for tumor cell proliferation and survival. They include ALK fusions such as NPM-ALK and EML4-ALK. Inhibiting the chaperone function of Hsp90 therefore represents an alternative strategy to kinase inhibition in ALK-driven tumors. The efficacy of Hsp90 inhibition in overcoming ALK TKI resistance has been demonstrated in preclinical models of NSCLC [107]. Moreover, a therapy combining Hsp90 and ALK inhibition could delay the emergence of resistance to TKI [108]. In a clinical trial on NSCLC, the Hsp90 inhibitor ganetespib overcame multiple forms of crizotinib resistance in molecularly selected ALK-rearranged patients. However, side effects of Hsp90 inhibitors are usually worse than those of second-generation ALK-inhibitors, which makes them difficult to use [80,109].

### 5.4. Association of ALK TKI and Inhibitors of Bypass Signaling Pathways

As mentioned above, several signaling pathways can bypass the tumor cells’ addiction to ALK. Therefore it seems rational to use ALK TKI in combination with therapy targeting the most common pathways activated in cancer. This strategy has been successful in vitro and in vivo in targeting the Ras/MAPK, EGFR, and IGF-1R kinases in ALK fusion-positive lung cancer, but is not yet applied in the clinic [81,85,110].

## 6. Conclusions

To summarize, ALK is a receptor tyrosine kinase involved in many types of tumors. It is an attractive molecular target, and the development of successive generations of ALK TKI has led to significant clinical improvements, especially for patients with ALK-positive NSCLC. However, resistance to TKI treatment invariably occurs, leading to relapse and tumor dissemination. Some mechanisms of resistance have been elucidated, and can be specifically addressed by targeted therapies in order to overcome the treatment failures. We have described a promising strategy taking advantage of the proapoptotic properties of ALK-derived peptides for ALK-dependent tumor cells which enhances the cytotoxic effect of an ALK TKI. Other alternative strategies target signaling pathways downstream of ALK, bypass ALK, or favor its degradation. Most importantly, as indicated by experimental studies, future clinical improvement for patients bearing ALK-positive tumors will likely stem from associating an ALK TKI with one of these alternate therapies, bringing synergy to kill tumor cells.

## Figures and Tables

**Figure 1 cancers-11-00275-f001:**
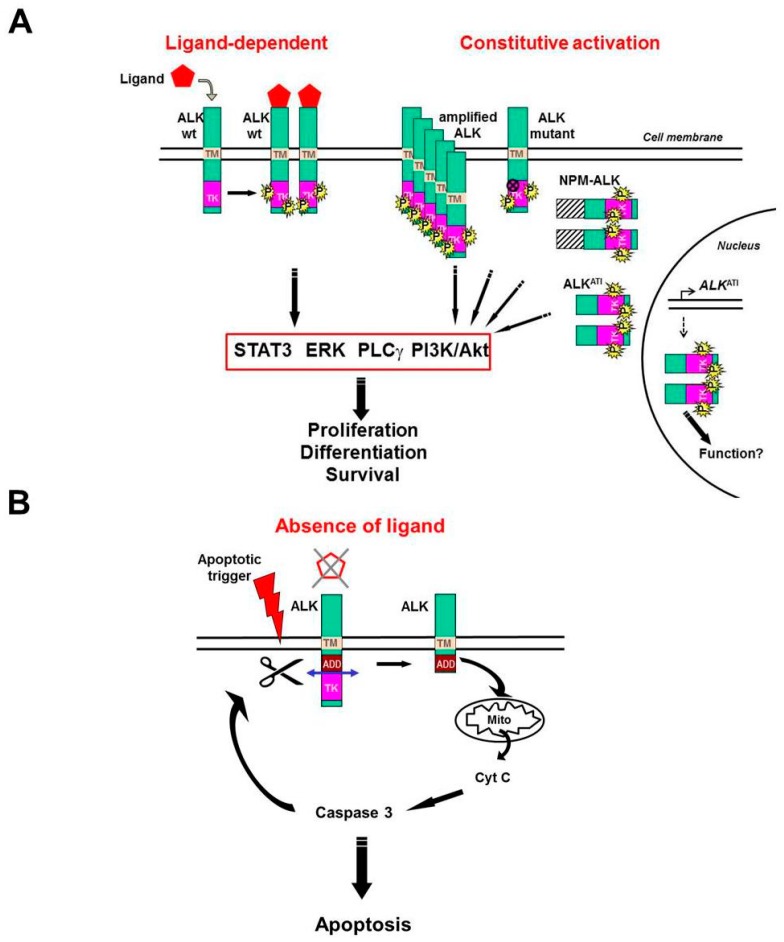
Model for Anaplastic Lymphoma Kinase (ALK) positive and negative signaling. ALK receptor activation requires homodimerization and transphosphorylation of its tyrosine kinase domain. (**A**) ALK activation can be achieved in the presence of a ligand (left), or constitutively (right) when *ALK* is either amplified, mutated, involved in a fusion such as NPM-ALK, or a product of *ALK* alternative initiation of transcription (ALK^ATI^). Note that the subcellular localization of all these ALK isoforms is different: the wild type or mutant full length receptor is localized at the cell membrane, whereas NPM-ALK and ALK^ATI^ are intracellular, as they lack the extracellular and transmembrane domains of ALK. Both NPM-ALK and ALK^ATI^ also localize to the nucleus (and nucleolus for NPM-ALK), however, only ALK^ATI^ homodimerizes and is activated in this compartment, and therefore is represented on this schema. Its function in the nucleus needs to be clarified, although it is reported to induce chromatin modifications. Signaling triggered by activated ALK includes the STAT3, ERK, PLCγ, and PI3K/Akt pathways leading to cell proliferation, differentiation, and survival. (**B**) In the absence of ligand, the ALK receptor promotes apoptosis via caspase 3 activation through mitochondrial release of cytochrome C. In this state, the kinase is inactive and the receptor likely monomeric. Inactive ALK (non-ligated or kinase inactive) is cleaved by caspase 3, thus exposing the ADD intracellular domain (upstream of the cleavage site) and amplifying apoptosis in a positive feedback loop. TM: transmembrane; TK: tyrosine kinase; P: phosphorylation on tyrosine residues; ADD: addiction/dependence domain; Mito: mitochondria; Cyt C: cytochrome C.

**Figure 2 cancers-11-00275-f002:**
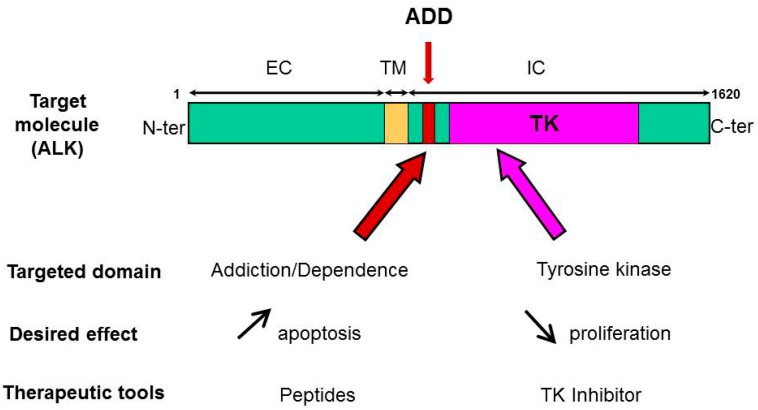
ALK targeting at two fronts. Targeting ALK tyrosine kinase domain with a specific inhibitor can decrease tumor cell proliferation and induce apoptosis. On the other hand, ALK-derived peptides mimicking the ADD domain of ALK were found to be proapoptotic. A combination treatment can bring additive or synergistic efficiency to kill ALK-dependent tumors.

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
