# Peer review of "Targeting ALK in Cancer: Therapeutic Potential of Proapoptotic Peptides"

_cancers, 2019, doi:10.3390/cancers11030275_

Round 1
Reviewer 1 Report
This is an excellent review on the role of proapoptotic peptides as a mediator of ALK signaling inhibition written by authors who have also conducted original research in the field
The authors are to be congratulated for their clarity and comprehensive cover of the topic on this very informative review. The only comment I would like to make is that in the session of targeting of ALK-dependent pathways, the authors should add a small paragraph regarding the secondary ALK mutations conferring resistance to 1st -generation ALK inhibitors (crizotinibn) discovered in NSCLC and the role of the 2nd (ceritinib, alectinib, brigatinib) and the 3rd generation (lorlatinib) ALK inhibitors as potent agents that can help to overcome resistance in ALK positive NSCLC
Author Response
Section 3 “ALK tyrosine kinase inhibitors” has been enriched, and divided in 2 subsections distinguishing first and next generation inhibitors.
We also quote a recent review published in Cancers in 2018 by Sharma et al:
“A recent review by Sharma et al listed the different ALK TKI, their specificity for other kinases, the (completed or ongoing) clinical trials currently testing these inhibitors and their observed adverse effects (Sharma et al, Cancers 2018). Another study focused more specifically on clinical trials of ALK inhibitors in neuroblastoma (Janoueix-Lerosey et al, Cell Tissue Res 2018).”
Reviewer 2 Report
The authors provided a comprehensive review on ALK that is an oncogenic receptor tyrosine kinase expressed in several cancers. The review is very informative and is compiled meticulously with distilled assessment of the field. My minor suggestion would be to included the clinical trials (ongoing and in pipeline) in the tabular format that are based on ALK as a molecular target.
Author Response
Section 3 “ALK tyrosine kinase inhibitors” has been enriched, and divided in 2 subsections distinguishing first and next generation inhibitors.
We chose not to include the table suggested by reviewer 2, because such a table has been published in a recent review by Sharma et al (Cancers 2018), that we quote in our text. We also quote another study focused more specifically on clinical trials of ALK inhibitors in neuroblastoma (Janoueix-Lerosey et al, Cell Tissue Res 2018), which also contains a table.
Reviewer 3 Report
Brief summary:
Several ALK tyrosine kinase inhibitors have been developed and used to treat ALK-driven tumors. However, drug resistance eventually develops, especially in ALK-rearranged NSCLC, with different mechanisms, highlighting an urgent need for new therapeutic stratigies like combination therapy. Based on their previous research findings, the authors here in this review discuss the therapeutic potential of proapoptotic ALK-derived peptides which might offer a novel therapeutic option for treating ALK-driven tumors.
Broad comments:
Starting with a detailed introduction about the discovery of ALK and its ligands as well as their physiological functions in zebrafish model, a brief summary of ALK signaling and ALK inhibitors which have been intensively reviewed elsewhere, the authors extend this review with a focus on ALK proapoptotic signaling and the therapeutic potential of proapoptotic ALK-derived peptides, as well as other possible combination strategies to overcome ALK TKI drug resistance.
The finding of ALK as a dependence receptor which under specific conditions can trigger/enhance apoptosis is very interesting, however, there are no comprehensive and deep studies to clarify the underlying mechanisms. In addition, the authors have found that ALK ADD derived peptide P36 can somewhat inhibit growth of ALK-positive ALCL and NB cells and offer additive efficacy when combined with ALK inhibitor, however, there is no any preclinical in vivo data so far to support the potential use of this peptide. Though based on an interactome study, the authors proposed a p53-dependent apoptosis model, this is not convincing and scientific hypothesis is not enough to support a translational use.
Compared to ALK-positive NSCLC which usually affects adults, the pediatric ALCL shows better response to ALK TKIs and less drug resistance. A recent publication in CSH Molecular Case Studies also shows a complete response to the second generation ALK TKI ceritinib in a high-risk NB patient with ALK I1171T activating mutation and 11q deletion, and no relapse during and after around 3 years ceritinib monotherapy. The authors also mentioned that drug resistance occurs more frequently in ALK-positive NSCLC, however, there is no research data of P36 peptide on such lung cancer cells which might be more useful.
Taken together, it would be interesting to see a translational use of this kind peptide in the future, but the authors need perform more studies to get enough solid data before summarize them in a review paper.
Specific comments:
1. line 21 in the ’Abstract’ and line 44 in the ’Introduction’ and line 360 in ’conclusion’, the authors wrote ’ALK is an oncogenic receptor tyrosine kinase…’. This sentence might be misleading because ALK by itself is an RTK whose activation is dependent on ligand binding, therefore it is not ongenic. Only aberrant activation of ALK kinase either by GOF point mutations, amplification or fusion with dimerizing partners is oncogenic.
2. line 24 ’ Since ALK–expressing tumor cells are dependent…’. Actually many neuroblastoma cells express ALK, but they don't rely on ALK activity. Use ’ALK-driven tumor cells’ might be more proper.
3. line 62-64, for the description ’ Other solid tumors expressing the full length ALK receptor include…’, should provide a reference here.
4. line 89 ’These secreted molecules have been named ALK and LTK ligands (ALKAL)’. Better to add ’approved by HUGO Gene Nomenclature Committee’. Then readers know why they have several different names/symbols.
5. line 111 ’Ligand binding to ALK receptor induces cell proliferation and/or differentiation in neuroblastoma cells while triggering the ERK pathway’. In NB cells, ALK signals through PI3K-AKT-ERK5 pathway to promote cell proliferation (see references: Umpathy G. et al, Sci Signal. 2014 Oct 28;7(349) The kinase ALK stimulates the kinase ERK5 to promote the expression of the oncogene MYCN in neuroblastoma; Umpathy G. et al, Sci Signal. 2017 Nov 28;10(507). MEK inhibitor trametinib does not prevent the growth of anaplastic lymphoma kinase (ALK)-addicted neuroblastomas.)
6. for fig 1 (model of ALK signaling…), first, the canonical ligand-dependent ALK activation leads to cell differentiation based on studies in Drosophila and zebrafish models, and the ALK oncogenic activation leads to proliferation and survival. Second, the constitutive activation of ALK should also include ALK activating point mutations and ALK amplication, not just ALK fusions.
7. line 159 ’ while it is not present in normal tissues’ is different from description in line 58-59 ’ ALK is mainly expressed in the central and peripheral nervous system during development and at lower levels in the adult’.
8. line 171 ’ approved by the FDA for treating patients also inhibits the MET kinase’, actually crizotinib is know to inhibit ALK, MET and ROS1.
9. line 321, 5.2 Reactivation of p53 with an MDM2 inhibitor. Study (Hata AN et al, Oncogene. 2017 Nov 23;36(47):6581-6591. Synergistic activity and heterogeneous acquired resistance of combined MDM2 and MEK inhibition in KRAS mutant cancers.) has demonstrated that reactivation of wild type p53 could increase the risk of mutating p53, resulting in even worse situations.
Author Response
Thanks for your comments. Please see the responses in the attached document.

Reviewer 4 Report
The review is very well written. The authors have done a comprehensive review of the function of ALK in development and disease and most importantly they provide alternative strategies of targeting ALK and describe the mechanism by which these strategies work to tackle TKI resistance. The paper does not need any further changes and i highly recommend for publication.
Author Response
Reviewer 4 was satisfied with the paper and did not ask for any change
Round 2
Reviewer 3 Report
Now I recommend it for publication.